# Relationship Between Coronal Plane Alignment of the Knee Phenotypes and Distal Femoral Rotation

**DOI:** 10.3390/jcm14051679

**Published:** 2025-03-01

**Authors:** Vicente J. León-Muñoz, José Hurtado-Avilés, Fernando Santonja-Medina, Francisco Lajara-Marco, Mirian López-López, Joaquín Moya-Angeler

**Affiliations:** 1Department of Orthopaedic Surgery and Traumatology, Hospital General Universitario Reina Sofía, Avda. Intendente Jorge Palacios, 1, 30003 Murcia, Spain; drlajaramarco@gmail.com (F.L.-M.); jmoyaangeler@gmail.com (J.M.-A.); 2Instituto de Cirugía Avanzada de la Rodilla (ICAR), C. Barítono Marcos Redondo 1, 30005 Murcia, Spain; 3Department of Surgery, Paediatrics and Obstetrics & Gynaecology (Faculty of Medicine), Avda. Buenavista 32, El Palmar, 30120 Murcia, Spain; fernando@santonjatrauma.es; 4Sports & Musculoskeletal System Research Group (RAQUIS), Faculty of Medicine, University of Murcia, Avda. Buenavista 32, El Palmar, 30120 Murcia, Spain; joseaviles@um.es; 5Department of Orthopaedic Surgery and Traumatology, Hospital Clínico Universitario Virgen de la Arrixaca, Ctra. Madrid-Cartagena, s/n, El Palmar, 30120 Murcia, Spain; 6Servicio de Coordinación y Aplicaciones Informáticas, Subdirección General de Tecnologías de la Información (Servicio Murciano de Salud), C. Central, 7, Espinardo, 30100 Murcia, Spain; miriam.lopez5@carm.es

**Keywords:** knee phenotypes, coronal plane alignment of the knee (CPAK), distal femoral rotation, condylar twist angle, knee replacement, knee arthroplasty, knee alignment technique

## Abstract

**Background:** The coronal plane alignment of the knee (CPAK) classification categorises nine phenotypes based on constitutional limb alignment and joint line obliquity and can be used in healthy and arthritic knees. In total knee arthroplasty surgery, some morphological variables in planes other than the coronal plane are particularly interesting. One example is the distal femoral rotation. Our study aimed to search for relationships between phenotypes based on CPAK classification and distal femoral rotation. **Methods:** Data from 622 cases in 535 osteoarthritic patients who underwent primary total knee arthroplasty were retrospectively analysed. Computed tomography imaging was employed to ascertain the mechanical lateral distal femoral angle, the mechanical medial proximal tibial angle, and the distal femoral rotation (quantified using the condylar twist angle). **Results:** The variables were perfectly uncorrelated according to the regression equations, with a Coefficient of Determination of 0.0608 for the condylar twist angle. Upon visualising the condylar twist angle function using a contour map or surface curves with low interpolation, it became evident that the data did not follow any discernible pattern. Employing ANOVA, we found some statistically significant differences between the distributions of the CPAK groups for the condylar twist angle (F = 5.81; *p* < 0.001). **Conclusions:** Our study found no relevant relationships between coronal plane alignment, according to the CPAK classification, and the distal femoral rotation in the sample population studied. Perhaps the stratification of the CPAK groups (i.e., a purely arithmetical aspect) hides possible relationships between the coronal and the axial planes.

## 1. Introduction

The coronal plane alignment of the knee (CPAK) classification reported by MacDessi et al. in 2021 provides a simple and pragmatic distribution of nine phenotypes for coronal knee alignment [1]. The CPAK classification is based on constitutional limb alignment and joint line obliquity and can be used in healthy and arthritic knees [1]. It is important to note that the CPAK classification is not the only classification proposed for the coronal plane of the knee [2,3,4,5,6,7,8]. Furthermore, not all authors agree that the CPAK classification helps stratify the different phenotypes in the coronal plane [9,10,11,12]. In addition to the limitation of being reduced to a phenotypic description of the knee in a single plane (coronal), various arguments have been published against the CPAK classification adequacy, including the following: (1) The three-dimensional orientation of the articular surface relative to the floor does not correlate with the two-dimensional orientation of the coronal joint line, and the CPAK classification types do not correctly represent it [11]. (2) The CPAK matrix groups do not show a direct correlation with a specific pattern of extra-articular deformity [10]. (3) The CPAK joint line obliquity measurement technique can be misleading in defining the position of the apex of the knee joint line obliquity, and the agreement between them is less than 50% [9]. In addition, in a recent publication from our group [13], we observed something that has been reported in several published studies analysing the CPAK classification of different populations: the low percentage representation of CPAK groups VII and, essentially, VIII and IX, with percentages of less than 1% in most studies [1,14,15,16,17,18,19,20,21]. 

The relationship of the different morphological or angular variables between the three planes is an area for improvement in any attempt to phenotypically classify the knee joint. Regardless of the alignment strategy chosen for total knee arthroplasty (TKA) surgery, including mechanical, adjusted mechanical, anatomical, unrestricted, or restricted kinematic, inverse kinematic, or functional alignment, it is a mistake to focus solely on the coronal plane and ignore the various variables in the sagittal and axial planes that can potentially affect the outcomes of the operation.

Several authors have previously investigated the relationship between the coronal plane and variables in the other two planes [22,23,24,25,26]. We also observed a linear relationship between the coronal alignment, the rotational geometry of the distal femur, and the tibial torsion [27]. More recently, other authors have investigated the relationship between phenotypes based on the CPAK classification and sagittal and axial plane geometry [28,29,30]. Contrary to Corbett et al., who claimed that the CPAK phenotype has little correlation with three-dimensional alignment characteristics and that these results do not support the need to extend the CPAK classification beyond coronal plane alignment [28], Ziegenhorn et al. postulated that there is a correlation between the coronal alignment of the lower limb and femoral torsion and that therefore, this may provide a basis for extending the CPAK classification beyond the coronal plane [29]. Jagota et al. [30] also found no clear relationship between variables in the sagittal and axial planes and the coronal description according to the CPAK classification criteria.

In TKA surgery, some morphological variables in the axial and sagittal planes are particularly interesting. For example, distal femoral rotation can condition the rotational alignment of the femoral component. It has been widely documented that the rotational misalignment of the femoral component can result in suboptimal outcomes, including discomfort, pain, patellar maltracking, instability, stiffness, inadequate gait kinematics, and arthrofibrosis [31,32,33]. It can also reduce prosthetic survival. However, the optimal rotational position of the femoral component to avoid these adverse outcomes remains a subject of ongoing debate and has not yet been fully resolved. To date, a definitive correlation between the rotational position of the femoral component and the ensuing clinical and functional outcomes remains firmly established, underscoring the necessity for a patient-centred approach that considers individuality.

The important clinical implications of achieving an adequate distal femoral rotation after TKA surgery motivated the present study. It aimed to evaluate whether there is an association between coronal alignment according to the CPAK classification and the distal femoral rotation in a population sample with osteoarthritis (OA).

## 2. Materials and Methods

This study was retrospective, cross-sectional and observational. Data from 622 cases in 535 OA patients who underwent primary TKA were retrospectively analysed. In all cases, these were patients undergoing TKA surgery and, therefore, Kellgren–Lawrence grade 3 or 4 knees. Our study group, with an average age of 70.3 years (range 45 to 84) and an average body mass index of 29.08 Kg/m^2^ (range 22.8 to 41.8), comprised 364 (58.5%) female and 258 (41.5%) male cases. Out of an initial series of 709 knees, we excluded cases that, due to previous interventions (e.g., femoral or tibial osteotomies) or fractures, presented a possible alteration of the native constitutional axes (23 cases), patients with a flexion contracture of 15 degrees or more (because the study protocol for computed tomography (CT) scanning states that measurement accuracy may be lost in knees with a severe flexion contracture; 17 cases) and patients with ipsilateral hip replacements (as the implantation of the hip prosthesis may alter the constitutionality of the knee alignment [34,35]; 47 cases).

The methodology employed in this study is consistent with the approach utilised in other publications by our research group [36,37]. We used the Somatom Scope scanner (Siemens Healthcare GmbH, Erlangen, Germany) for image acquisition. For the CT scan, images were acquired with the patient positioned supine in the isocenter of the gantry with the leg of interest at full extension. The acquisition comprised three brief spiral axial scans encompassing the hip, knee, and ankle regions. Each acquisition was accurately centred and zoomed to ensure that the field of view (FoV) circumscribed the region of interest to the greatest possible extent. The scans were acquired in slices of at least 512 x 512 pixels. The thickness of a single slice was 2 mm for the hip and ankle and 0.6 mm for the knee, with a maximum field of view (FoV) of 200 mm. The peak voltage was set to 130 kV, and the X-ray tube current was set to 60 mA. The average effective radiation dose per CT scan was 0.4 mSv. The images were archived on the Picture Archiving and Communication System (PACS, Siemens Healthcare GmbH, Erlangen, Germany) server in the international standard DICOM (Digital Imaging and Communications in Medicine) format. Measurements on CT scans were performed using MyPlanner^®^ v4.2.41 software (Medacta International, Castel San Pietro, Switzerland) to create virtual 3D models by the MyKnee engineering staff.

The alignment was measured, and the value obtained was rounded to 0.5 degrees. The non-weight-bearing mechanical hip–knee–ankle angle (mHKA angle) was obtained from the intersection of the femoral and tibial mechanical axes. The mechanical lateral distal femoral angle (LDFA) was defined as the lateral angle between the femoral mechanical axis and the distal femoral joint line (the tangent between the most distal points of the femoral condyles). The medial mechanical proximal tibial angle (MPTA) was defined as the medial angle between the mechanical axis of the tibia and the proximal tibial joint line (the line between the deepest points of the medial and lateral tibial condyles).

Distal femoral rotation was measured using the condylar twist angle (CTA) first described by Yoshioka et al. [38]. The CTA (Figure 1) is defined as the angle between the posterior condylar line (PCL) (the line connecting the most posterior margins of the lateral and medial posterior condyles) and the clinical or anatomical trans-epicondylar axis (cTEA) (a line connecting the tip of the medial and lateral epicondylar prominences of the femur). 

The CPAK classification evaluates two criteria: constitutional limb alignment (or the arithmetic hip–knee–ankle angle (aHKA angle)) and joint line obliquity (JLO) [1]. These criteria can be calculated according to the LDFA and the MPTA. Constitutional limb alignment is described as varus, neutral, or valgus. The aHKA is calculated by subtracting the LDFA value from the MPTA value. The JLO is described as apex distal, neutral, or proximal and calculated by adding the value of MPTA to the value of LDFA. The three subgroups of aHKA are crossed with the three subgroups of JLO to yield the nine CPAK types [1].

Statistical analysis was performed using the Statistical Package for the Social Sciences (SPSS), version 25 for Windows (SPSS, Inc., Chicago, IL, USA), and Minitab Statistical Software, version 22 for Windows (Minitab LLC, State College, PA, USA). Our study followed the ethical standards of the World Medical Association Declaration of Helsinki, as revised in 2024. The institutional review board of the author’s institution approved the study protocol (CEIC-HMM-16/19). Given the research’s retrospective nature and the medical imaging’s anonymisation (employing a user-defined anonymisation function on the identifiers (the medical record number of each patient) in the Microsoft Excel workbook (Microsoft 365)), the institutional review board considered the study exempt from requiring patients’ informed consent.

## 3. Results

The values below Q1− (1.5 interquartile range (IQR)) and above Q3+ (1.5 IQR) for the three variables (not the individual outliers of each variable, as each pair of aHKA and JLO values must correspond to a CTA value) were identified. The values mentioned above, which we classified as outliers, were removed from each distribution. This step is of paramount importance, as it prevents the inclusion of patients whose bone morphology is more attributable to significant arthritic bone loss than to the phenotypic characteristics. The IQT method with a scale of 1.5 is the most widely used, and there is a certain consensus in the statistical literature that it should be adopted as the standardisation method, removing outlier data that are at least 2.7 standard deviations away from the arithmetical mean [39,40].

We calculated the plane that best fit each distribution, as shown in Figure 2.

The variables were perfectly uncorrelated according to the regression equation: CTA = −4.45 − 0.0897 aHKA + 0.0540 JLO (r^2^ = 0.0608). Given the absence of correlation, we obtained a graphical representation that can provide information on the characteristics of the distributions. 

The distribution of the response variables was evaluated (see Figure 3). None of the plans exhibited a discernible distribution pattern.

Upon visualising the CTA function using a contour map (which is a way to depict functions with a two-dimensional input and a one-dimensional output) formed by contour lines on the aHKA, JLO plane (see Figure 4), it became evident that the data do not follow any discernible pattern.

The distribution of the aHKA, JLO, and CTA was divided into nine CPAK classification groups, with single outliers discarded for each variable. Table 1 shows the values of the variables aHKA, JLO, and CTA according to the CPAK classification groups. Upon examination of the variables using normality tests (Kolmogorov–Smirnov and Shapiro–Wilk tests, depending on the sample size), it was determined that all distributions exhibited the requisite normality characteristics except for the CTA variable for the CPAK IV group. This is not a cause for concern, as this variable follows unimodal distribution and has a sufficiently large *n* (central limit theorem). 

Using ANOVA, we found that there were statistically significant differences between the distributions of the CPAK groups for CTA (F = 5.81; *p* < 0.001). We calculated Tukey’s comparisons to see which distributions differed statistically, as shown in Figure 5.

## 4. Discussion

Our study aimed to search for relationships between phenotypes based on the CPAK classification and the distal femoral rotation. There is no definitive assertion regarding the existence or absence of a relationship between the stratification according to the CPAK classification and variables on other planes, such as distal femoral rotation. In 2021, MacDessi et al. [1] published a simple and pragmatic classification system for coronal knee alignment based on constitutional limb alignment and joint line obliquity. This system, known as the CPAK classification, can be used to categorise nine phenotypes for coronal knee alignment in healthy and arthritic knees. The CPAK classification is one of several proposed for the coronal plane of the knee [2,3,4,5,6,7,8]. Its limitations have been criticised [9,10,11,12,19]. Possibly, the decision to set the boundaries for neutral aHKA at −2º and 2° and for neutral JLO at 177° and 183°, and thus, to set mean typical values of 0° for aHKA and 180° for JLO (which mathematically assumes a mean value of 90° of normality for LDFA and MPTA), is probably the origin of some of the limitations of the CPAK classification. Several publications, including the classic paper by Bellemans et al. [41] describing the “constitutional varus concept”, have already shown that the mean values of LDFA and MPTA differ from 90°. For example, in the present study of 622 knees (remembering that these are OA knees), the mean value (with standard deviation) for LDFA was 91.75° (3.08°) and for MPTA was 87.28° (2.91°). However, the CPAK classification is, nowadays, perhaps the most widely used categorisation to describe a grouped distribution of the different variants of coronal plane alignment [1,13,14,15,16,17,18,19,20,21,42,43,44]. For some authors, the current focus on the static constitutional alignment of the lower limb according to one of the many phenotype classifications may overlook dynamic alignment measures during gait. They advocate for the concept of dynamic HKA, which ultimately measures the coronal alignment of the knee throughout the gait cycle [45].

Corbett et al. [28] recently measured the coronal, sagittal and rotational alignment on the CT images of 509 OA knees. Their study aimed to determine if sagittal and rotational knee alignments vary among CPAK types. The authors found few clinically important differences in sagittal and rotational alignment between the CPAK types, suggesting that the CPAK phenotype has little correlation with three-dimensional (3D) alignment characteristics. In the opposite direction, Ziegenhorn et al. [29] found a correlation between coronal alignment of the lower limb and femoral torsion after measuring digital 3D reconstructions of 1000 EOS images of the legs of patients with knee pain. The authors found no correlation for tibial torsion. Jagota et al. [30] recently performed a retrospective analysis of a CT database (7450 knees). They observed weak linear correlations between the axial and sagittal measurements assessed and the aHKA and JLO. In the present study, our observations indicate the absence of a relationship between the values utilised for the stratification of the different groups of the CPAK classification (aHKA and JLO) and the axial plane value analysed (CTA). Consequently, our perspective aligns with that of Corbett et al. [28] while disagreeing with Ziegenhorn et al. [29]. We do not consider extending the CPAK classification with variables from planes other than the coronal as necessary or feasible. Instead, we propose re-examining the boundaries delineating each group within the CPAK classification as a potentially fruitful avenue of research.

In a prospective study, our group analysed CTA, proximal femoral version, and tibial torsion in 3D reconstructions from CT images of 385 osteoarthritic knees [27]. We found that as coronal alignment changed from varus to valgus, femoral external rotation increased (r = 0.217, *p* < 0.001), and external tibial torsion increased (r = 0.248, *p* < 0.001). No correlation was found between the global coronal alignment and the femoral version.

Aglietti et al. [46] first reported a linear relationship between the coronal alignment and the distal femoral external rotation measured through the posterior condylar angle (PCA). PCA is the angle between the PCL and the surgical trans-epicondylar axis (a line connecting the lateral epicondyle’s most prominent point with the medial sulcus’s deepest point). Luyckx et al. [22] identified a clear linear relationship between the overall coronal alignment and the rotational geometry of the distal femur consistent with Aglietti’s prior postulate. In the three studies mentioned above [22,27,46], the mechanical tibiofemoral angle (the angle between the femoral and tibial mechanical axes, or, in other words, the mHKA) was used as the reference for coronal alignment. However, we used both the aHKA and the JLO for our current comparison and did not observe any relationship in line with Corbett et al. [28]. We analysed possible correlations with both parametric (based on the normality indicated by graphical methods) and non-parametric (based on the results of the Kolmogorov–Smirnov test for variables < 0.05) tests in our series using the non-weight-bearing mHKA value and observed a significant relationship (Pearson: r = 0.249, *p* < 0.001; Spearman: r = 0.236, *p* < 0.001) between the mHKA and distal femoral rotation as measured by CTA. Consistent with previous studies [22,27,46], distal femoral external rotation increases when the valgus alignment increases. These results force us to interpret the presence or absence of a relationship between the coronal plane alignment and distal femoral rotation as dependent on the mathematical terms used. The result is different when using non-weight-bearing mHKA or aHKA, so we can infer that non-weight-bearing mHKA (and perhaps the weight-bearing mHKA, although we cannot say for sure) and aHKA do not measure exactly the same thing. 

The existing difference between the distributions of the CPAK groups for CTA (F = 5.81; *p* < 0.001) does not imply a correlation (i.e., a dependent relationship between the CTA distribution data and the variables that define each CPAK classification group in consideration of the regression equations obtained). Both concepts are not mutually exclusive.

There are several potential limitations to this study. First, we only classified the cases according to the CPAK classification. We did not contrast the cases with the functional knee phenotype concept [2,3,4,47] or the classifications proposed by Lin et al. [5], Mullaji et al. [7,8], or Marchand et al. [6]. Second, this was a retrospective cohort study based on the database of a single centre of patients originating from a determined region of our country, which limits the generalisability of the results to a larger or broader population. Third, this study only included data from patients with knee OA (Kellgren–Lawrence grade three or four) and did not examine healthy patients. However, some studies argue that the CPAK classification does not change with OA progression [17,48], and most morphological analyses are based on CT scans performed in the context of preoperative planning for TKA surgery (and, therefore, late-stage OA). Fourth, we did not analyse the values regarding the femoral version, femoral bowing, tibial torsion, and rotation at the knee. Fifth, we removed values outside the interquartile range, as we considered these values to be outliers in the data set and incapable of reflecting the natural variability of the data. However, this would likely have a limited effect on the results, given the large numbers in this cohort. Eliminating outliers is imperative in any study because they impact the normality of the sample and influence the sample mean and standard deviation. Sixth, we evaluated the angular values (mHKA angle, LDFA, MPTA, and CTA), and thus, aHKA and JLO were obtained under non-weight-bearing conditions from the CT scan studies. Notwithstanding this, recent observations [49] posit that CT scan-based measurements are more accurate than full-length anteroposterior lower limb radiographs. Seventh, although our study analysed 622 knees to minimise type I statistical errors, the numbers of cases grouped into CPAK types VII to IX were small. We do not consider this a significant limitation, as these patients are rare in the general population based on the CPAK classification. Finally, we lack a specific determination of intra- and inter-observer variability. However, based on the findings of earlier studies [24], these variabilities can be assumed to be negligible.

## 5. Conclusions

We did not observe any correlation between the coronal plane alignment according to the CPAK classification groups with distal femoral rotation (CTA). On the other hand, when we used the mHKA, and not the aHKA, for the analysis, there was a correlation: the more significant the valgus, the greater the distal femoral external rotation. Perhaps the stratification of the CPAK groups (i.e., a purely arithmetical aspect) hides possible relationships between the coronal and axial planes. When using the CPAK classification in total knee arthroplasty surgery planning, we must pay close attention to the morphological and angular characteristics in the coronal plane and, independently, to these characteristics in the axial plane.

## Figures and Tables

**Figure 1 jcm-14-01679-f001:**
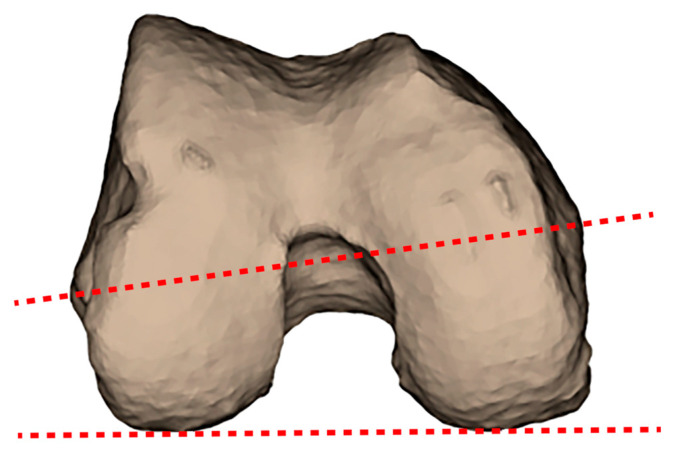
The condylar twist angle (CTA) is the angle between the posterior condylar line (a line connecting the most posterior margins of the lateral and medial posterior condyles) and the clinical or anatomical trans-epicondylar axis (a line connecting the tip of the medial and lateral epicondylar prominences of the femur).

**Figure 2 jcm-14-01679-f002:**
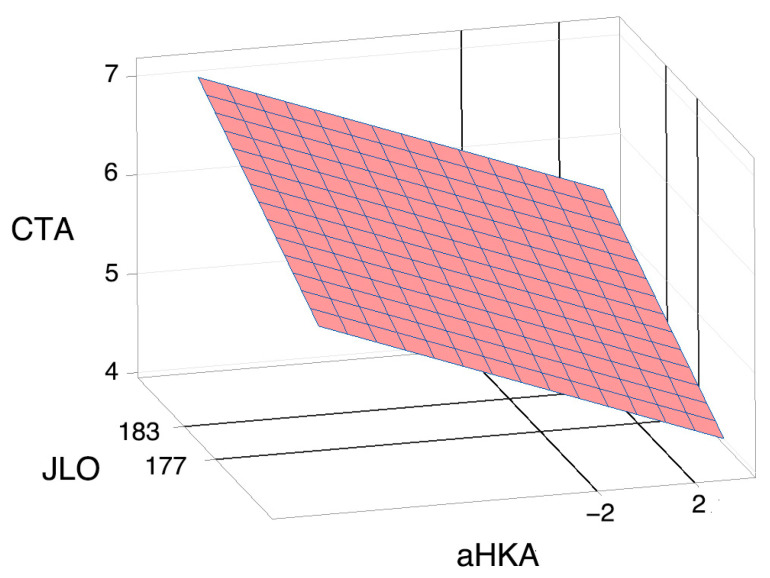
The plane adjusted by the least squares method to the values of the condylar twist angle (CTA), resulting from the functions of two variables, f(aHKA, JLO) = CTA, where aHKA is the arithmetic hip–knee–ankle angle and JLO is the joint line obliquity. Values are in degrees. By representing an actual discrete function of a real variable (f: R→R) on a plane defined by a Cartesian coordinate system, it is possible to fit a straight line to its values using the least squares method. In the diagram, a three-dimensional coordinate space (R^3^) is represented, and in a similar way to the regression line for f: R→R, the graph shows the plane adjusted by the method of least squares to the values of an actual discrete function of two real variables (f: R^2^→R). The image represents the absence of correlation between the independent variables of f: R^2^→R.

**Figure 3 jcm-14-01679-f003:**
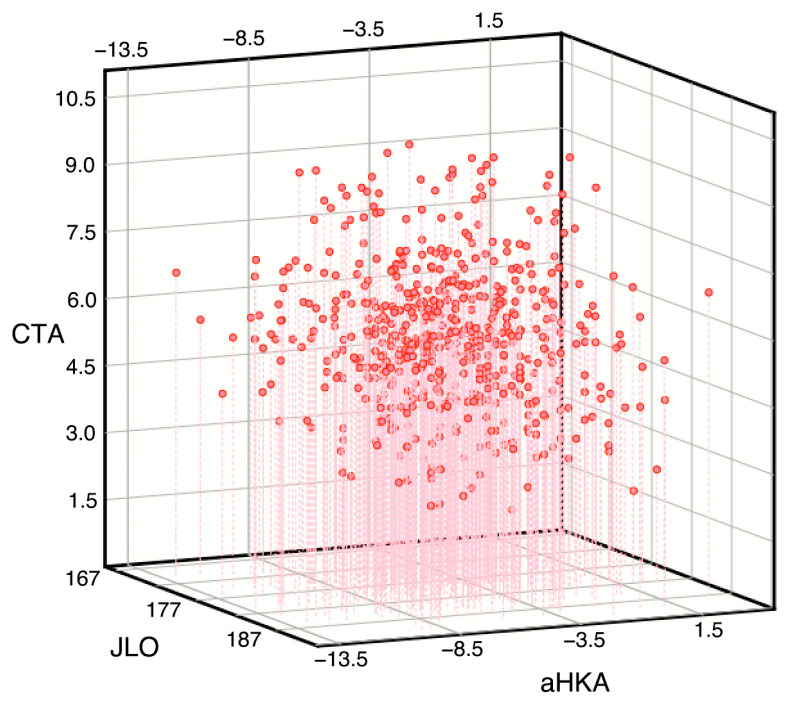
Distribution of point clouds resulting from the function of two variables, f(aHKA, JLO) = CTA, in the CTA-aHKA and CTA-JLO planes. aHKA: arithmetic hip–knee–ankle angle. JLO: joint line obliquity. CTA: condylar twist angle. Values in degrees. The figures represent the three-dimensional scatter plot obtained with the function f: R^2^→R in a manner analogous to its presentation on a two-dimensional plane for the function f: R→R. As is evident, the scatter plot does not exhibit any discernible pattern.

**Figure 4 jcm-14-01679-f004:**
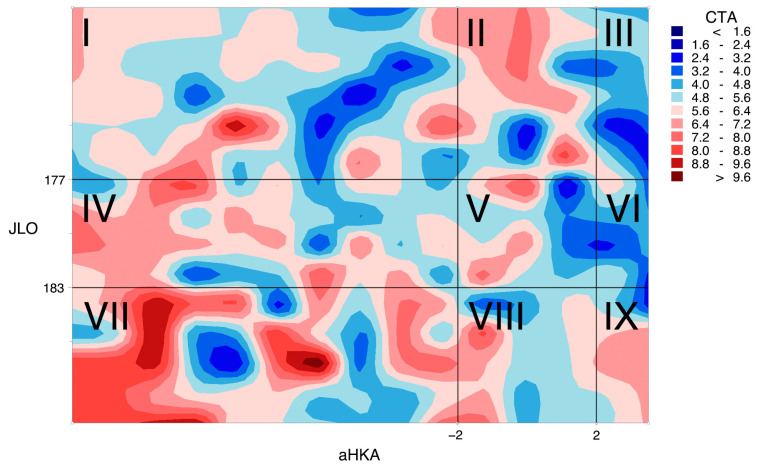
The following contour maps illustrate the distribution of CTA values in the f(aHKA, JLO) plane. In this plane, CTA represents the distal femoral rotation, aHKA represents the arithmetic hip–knee–ankle angle, and JLO represents the joint line obliquity. The Roman numerals correspond with the nine morphotypes of the Coronal Plane Alignment of the Knee (CPAK) classification. The distribution of values for CTA in each section of the plane is demonstrated (the red scale indicates high values of CTA, and the blue scale indicates low values of CTA). Values are in degrees. The image clearly represents the heterogeneous distribution of CTA values on the JLO-HKA plane, that is, of f: R^2^→R.

**Figure 5 jcm-14-01679-f005:**
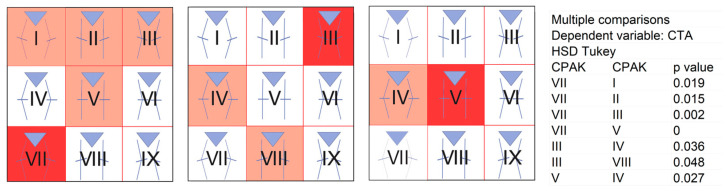
According to the Coronal Plane Alignment of the Knee (CPAK) classification, statistical differences existed between the groups for the variable condylar twist angle (CTA). Significant differences were observed between group VII and groups I, II, III and V, between group III and groups IV and VIII, and between group V and group IV. HSD Tukey, Tukey’s Honestly-Significant-Difference test.

**Table 1 jcm-14-01679-t001:** Values of the variables aHKA, JLO, and CTA according to the CPAK classification groups.

CPAK	aHKA (°)	SEM	CI95%	*n*	JLO (°)	SEM	CI95%	*n*	CTA (°)	SEM	CI95%	*n*
**I**	−5.57 (2.43)	0.21	−5.98/−5.17	137	174.26 (1.92)	0.16	173.94/174.58	138	5.48 (1.42)	0.12	5.25/5.72	137
**II**	−0.61 (1.13)	0.20	−1/−0.23	33	174.9 (1.41)	0.25	174.41/175.39	31	5.12 (1.53)	0.27	4.6/5.64	33
**III**	2.75 (0.59)	0.19	2.38/3.12	10	172.45 (3.01)	0.95	170.58/174.32	10	4.2 (1.01)	0.32	3.58/4.82	10
**IV**	−5.58 (2.24)	0.15	−5.87/−5.29	229	179.56 (1.61)	0.11	179.35/179.76	231	5.72 (1.25)	0.09	5.55/5.88	217
**V**	−0.58 (1.21)	0.17	−0.91/−0.25	51	179.37 (1.82)	0.26	178.87/179.87	51	4.96 (1.57)	0.22	4.53/5.39	50
**VI**	2.78 (0.48)	0.20	2.39/3.17	6	181.18 (0.88)	0.36	180.47/181.88	6	4.26 (1.84)	0.75	2.79/5.74	6
**VII**	−6.55 (2.61)	0.26	−7.07/−6.04	97	185.21 (1.73)	0.18	184.87/185.56	97	6.15 (1.79)	0.18	5.79/6.5	97
**VIII**	−0.67 (1.15)	0.32	−1.3/−0.04	13	185.64 (2.48)	0.69	184.29/186.99	13	6.12 (1.73)	0.48	5.18/7.05	13
**IX**	3.4 (0)	0.00	3.4/3.4	1	187.3 (0)	0.00	187.3/187.3	1	6.8 (0)	0.00	6.8/6.8	1

Data are presented as the mean (standard deviation), standard error of the mean (SEM), and confidence intervals (CI 95%). CPAK, Coronal Plane Alignment of the Knee classification; aHKA, arithmetic hip–knee–ankle angle; JLO, joint line obliquity; CTA, condylar twist angle.

## Data Availability

The raw data supporting the conclusions of this article will be made available by the authors upon request.

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
