# Peer review of "Relationship Between Coronal Plane Alignment of the Knee Phenotypes and Distal Femoral Rotation"

_jcm, 2025, doi:10.3390/jcm14051679_

Round 1

Reviewer 1 Report

Comments and Suggestions for Authors

I read this manuscript with interest. The study is well-structured and addresses an important topic, but some aspects need clarification, particularly regarding its novelty and data interpretation. Below are my comments for improvement.

  1. Novelty is unclear. The lack of correlation between CPAK classification and CTA has already been reported in previous studies【28, 30】. The authors should clarify what new knowledge this study adds. If the novelty lies in differences in patient population, methodology, or implications for TKA planning, these aspects should be explicitly stated. Simply confirming a known lack of correlation does not provide strong scientific impact.

  1. The statement that CPAK is the “most widely used” classification is exaggerated. Since CPAK was introduced in 2021, it is relatively new, and other classification systems exist【2–8】. The authors should carefully compare CPAK with other classifications and justify its usage.

  1. In lines 162–167, the claims “this step is of paramount importance” and “most widely used” lack supporting references. If IQR is considered a standard method, citations should be provided. Additionally, the rationale for excluding outliers using IQR should be explained. The authors should compare results with and without outlier exclusion to assess its impact on conclusions.

  1. Figures 2–5 are difficult to interpret. Instead of helping data interpretation, these figures add confusion. The purpose of each figure should be clarified, or unnecessary figures should be removed. Figures 4–5, in particular, are not intuitive, and simpler visualizations might be more effective.

  1. Table 1 and Figure 6 should clarify the direction of differences. The description only states that significant differences exist between groups but does not specify whether CTA is higher or lower in each group. This should be clearly stated to enhance readability.

  1. The discussion on CPAK classification limitations is appropriate, but comparisons with other classification systems (e.g., Functional Knee Phenotype, Mullaji classification) are missing. The practical utility of CPAK should be critically evaluated in this context.

  1. Although an association between CTA and mHKA was observed, its clinical significance is not well discussed. The authors should explain how this finding impacts TKA planning to improve the study’s clinical relevance.

Author Response

Comments and Suggestions for Authors Reviewer 1

I read this manuscript with interest. The study is well-structured and addresses an important topic, but some aspects need clarification, particularly regarding its novelty and data interpretation. Below are my comments for improvement.

Thank you very much for your thorough review, comments and suggestions. We will try to incorporate the changes you propose to improve the quality of the article and make it suitable for publication. Thank you very much for your kind contribution.

I will provide a point-by-point response to your comments and objections and indicate the modifications I propose to the manuscript to incorporate your comments.

Novelty is unclear. The lack of correlation between CPAK classification and CTA has already been reported in previous studies【28, 30】. The authors should clarify what new knowledge this study adds. If the novelty lies in differences in patient population, methodology, or implications for TKA planning, these aspects should be explicitly stated. Simply confirming a known lack of correlation does not provide strong scientific impact.

There is no categorical statement of the absence of a relationship. Ziegenhorn et al. published significant differences between CPAK groups regarding femoral torsion. On the other hand, Corbett et al. and Jagota et al. reported the absence of such a relationship. These three studies (and only these three) deal with the relationships between planes when the sample is stratified according to CPAK criteria. We believe that our study provides relevant information, which confirms the postulates of Corbett and Jagota but introduces an aspect not mentioned to date (which we are currently investigating): we hypothesise that the limits that stratify the CPAK groups present a methodological error, and that the absence of relationships is due to the repercussion of this error in the arithmetic formulation.

An example that may contextualise the importance of our study is the recent Mark Coventry Award: Does Matching the Native Coronal Plane Alignment of the Knee (CPAK) Improve Outcomes in Primary Total Knee Arthroplasty? Signed by Kraus from Meneghini's group. This study concluded that “matching a patient’s native knee coronal alignment classified by CPAK was not predictive of PROMs. This supports prior research that suggests TKA outcomes are multifactorial and related to complex interactions between implant position in three dimensions as well as soft-tissue balance and kinematics”. In other words, the interactions and relationships between the three planes are important and are related to the outcome of replacement surgery.

Furthermore, the statistical analysis methodology we have used is novel, and its dissemination is an approach that should be extended to the scientific community and dedicated to the locomotor system.

We have added information regarding the importance we think the study has: “There is no definitive assertion regarding the existence or absence of a relationship between the stratification according to the CPAK classification and variables on other planes, such as distal femoral rotation”.

The statement that CPAK is the “most widely used” classification is exaggerated. Since CPAK was introduced in 2021, it is relatively new, and other classification systems exist【2–8】. The authors should carefully compare CPAK with other classifications and justify its usage.

It is true, as it is a statement definite or assertive. That is why we think our statement can be improved by adding the adverb “perhaps”. If we search PubMed exclusively with the term “CPAK”, we get 120 results. 65 in 2024 and only in 2025 so far, 13 articles. Let us look for the articles that cite the paper from MacDessi et al., “Coronal Plane Alignment of the Knee (CPAK) classification” (https://doi.org/10.1302/0301-620X.103B2.BJJ-2020-1050.R). We found 104 citations, despite it being a recent publication (from 2021), as you claim. Citations for reference 2 = 72, for reference 3 = 82, for reference 4 = 118, for reference 5 = 31, for reference 6 = 0, for reference 7 = 9 and for reference 8 = 4 citations. There are other classification systems, but perhaps CPAK is being used more because of its simplicity (which is partly responsible for its weak points). We do not intend to enter into the debate about whether it is better or worse. Our study aims not to compare the different ways of classifying knee phenotypes in the coronal plane. We are noting a growing interest in this way of classifying knees and that it is the one we have studied. In our daily clinical practice, we use it to (partly) decide the type of alignment we use in each case and for teaching purposes. For these reasons, we have used this classification and not others.

We remove the sentence: “Despite criticism, knee surgeons use the CPAK classification most widely to define the coronal plane’s different morphotypes, providing a common language among healthcare professionals.”

We add the adverb “perhaps” in the sentence: “However, the CPAK classification is, nowadays, perhaps the most widely used categorisation to describe a grouped distribution of the different variants of coronal plane alignment [1, 13, 14–21, 40–42].”

In lines 162–167, the claims “this step is of paramount importance” and “most widely used” lack supporting references. If IQR is considered a standard method, citations should be provided. Additionally, the rationale for excluding outliers using IQR should be explained. The authors should compare results with and without outlier exclusion to assess its impact on conclusions.

The detection of atypical data using the IQR is a statistical method that has been widely adopted and strongly supported in literature for a very long time. It is little used in medical literature and widely used in other branches of science. We include the classic references that support it:

Tukey, John W. Exploratory Data Analysis. Addison-Wesley. (1977). ISBN 0-201-07616-0.

Freedman, D., Pisani, R., Purves, R. Statistics (4th ed.). Norton & Company. (2007). ISBN 978-0393929720.

The interquartile range (IQR) is a measure of dispersion statistics employed to assess the variability in the intermediate range of a data set, excluding outliers or extremes (i.e. values that can distort the traditional range). In other words, the IQR is an indicator that allows us to approximate the variability of observations close to the median. The interquartile range is a more robust measure than the traditional range because it is based on the quartile values rather than the maximum and minimum values of the data set.

The advantages of the interquartile range include the following:

  • It is not affected by extreme or atypical values.
  • This is particularly beneficial in cases where the distribution is highly asymmetric, meaning that the majority of values do not align with the mean. In such scenarios, the interquartile range becomes a more effective measure of dispersion when compared to the range.
  • Additionally, the median is a more suitable measure of central tendency in asymmetric distributions. While the interquartile range is relatively straightforward to calculate, it is not as simple as the traditional range.
  • The result is measured in the same units as the data being analysed.

However, the following disadvantage is associated with the interquartile range:

  • It takes longer to calculate than the traditional range.

Given the study’s objective of achieving phenotypic characterisation, the exclusion of atypical values encompasses cases in which the value of aHKA, JLO, and CTA is contingent on a marked osteoarthritic pathological alteration and not on a phenotypic morphological characteristic. Consequently, excluding these atypical values will yield a more precise result.

The dispersion of data that does not conform to a perfectly symmetrical distribution cannot be summarised by a single number. Consequently, the standard deviation can only assist in distinguishing data that is a certain distance from the sample mean, subsequent to eliminating outliers, thereby rendering the distribution more normal (i.e., closer to the symmetrical distribution model followed by the data studied).

Figures 2–5 are difficult to interpret. Instead of helping data interpretation, these figures add confusion. The purpose of each figure should be clarified, or unnecessary figures should be removed. Figures 4–5, in particular, are not intuitive, and simpler visualizations might be more effective.

An attempt has been made to expand the explanation of the figures to make them more comprehensible. This visual presentation of the results highlights the absence of a relationship. Should this expansion fail to resolve the issue, eliminating some figures remains open. The figure that is considered fundamental is 4. In conjunction with Table 1, it is essential for the presentation of the study’s results.

We have added to the explanation in Figure 2: By representing an actual discrete function of a real variable (f: R→R) on a plane defined by a Cartesian coordinate system, it is possible to fit a straight line to its values using the least squares method. In the diagram, a three-dimensional coordinate space (R^3) is represented, and in a similar way to the regression line for f: R→R, the graph shows the plane adjusted by the method of least squares to the values of an actual discrete function of two real variables (f: R^2→R). The image represents the absence of correlation between the independent variables of f: R^2→R.

We have added to the explanation in Figure 3: The figures represent the three-dimensional scatter plot obtained with the function f: R²→R in a manner analogous to its presentation on a two-dimensional plane for a function f: R→R. As is evident, the scatter plot does not exhibit any discernible pattern.

We have added to the explanation in Figure 4: The image clearly represents the heterogeneous distribution of CTA values on the JLO-HKA plane, that is, of f: R^2→R.

Figure 5 is an alternative way of displaying the information in Figure 4. The heterogeneity of the values can be observed in three dimensions. We have decided to eliminate it to avoid confusion.

Table 1 and Figure 6 should clarify the direction of differences. The description only states that significant differences exist between groups but does not specify whether CTA is higher or lower in each group. This should be clearly stated to enhance readability.

In table 1 we show the average CTA value for each CPAK group (with its SD) and, therefore, it is clear how the value of CTA is higher or lower in each group. Furthermore, we show the standard error of the mean (SEM) values, that is to say, it is the value that quantifies how far the values deviate from the sample average. We have highlighted the CTA values in bold in the table to facilitate rapid and straightforward comprehension.

CPAK

CTA (°)

SEM

I

5.48 (1.42)

0.12

II

5.12 (1.53)

0.27

III

4.2 (1.01)

0.32

IV

5.72 (1.25)

0.09

V

4.96 (1.57)

0.22

VI

4.26 (1.84)

0.75

VII

6.15 (1.79)

0.18

VIII

6.12 (1.73)

0.48

IX

6.8 (0)

0.00

The discussion on CPAK classification limitations is appropriate, but comparisons with other classification systems (e.g., Functional Knee Phenotype, Mullaji classification) are missing. The practical utility of CPAK should be critically evaluated in this context.

As mentioned previously, our work aims not to compare the different classifications or evaluate the usefulness or lack of usefulness of the CPAK classification. This topic alone may be enough for an extensive study that goes far beyond the objective of our analysis. We only intend to evaluate whether group stratification follows a relationship pattern with a variable of the axial plane, such as the CTA. We have introduced “notes” of possible errors in the CPAK classification that could justify the absence of relationships between variables of different planes that are established when considering the mHKA and not the aHKA. However, this is still an unproven hypothesis and is not yet firm data that would help us quantify the practical usefulness of the CPAK. We want to remain equidistant from any attempt at phenotypic classification in the coronal plane of the knee. Although it is an excellent suggestion for a study, the critical analysis of the different alignment classification options in the coronal plane, comparing the advantages and disadvantages of each.

Although an association between CTA and mHKA was observed, its clinical significance is not well discussed. The authors should explain how this finding impacts TKA planning to improve the study’s clinical relevance.

It has been widely published that rotational misalignment of the femoral component leads to an unsatisfactory outcome, including discomfort, pain, patellar maltracking, instability, stiffness, inadequate gait kinematics, arthrofibrosis, and may reduce prosthetic survival. More controversial and still unanswered is the optimal rotation of the femoral component to avoid such unsatisfactory results. Several reports in the literature have evaluated the most common techniques used to adjust the axial alignment of the femoral component, with conflicting results. An exact correlation between the rotational position of the femoral component and the clinical and functional outcome has not yet been demonstrated, underlining the importance of patient individuality. Regardless of the angle chosen to assess femoral rotation, several reports have shown a correlation between coronal alignment and rotational geometry of the distal femur. External femoral rotation increases as the coronal alignment changes from varus to valgus (10.1016/j.arth.2017.10.038, 10.1016/j.arth.2018.07.022, 10.1007/s00167-012-2306-x, 10.1016/j.otsr.2018.04.032 and 10.1016/j.otsr.2019.07.005). Maintaining the patient’s constitutional distal femoral rotation is the best option (90% of my TKA surgeries are with kinematic alignment). However, if the option is mechanical alignment, the rotation of the femoral component cannot be the same for severe varus as for a “normal axis” or severe valgus. In this sense, it is important to consider that perhaps with greater valgus, the external rotation of the femoral component should be increased somewhat. We add this information to the manuscript.

Thank you for your thorough review. Thank you for the ideas to improve the current manuscript and for those that we will be able to put forward prospectively in research similar to the current one. Thank you very much.

Reviewer 2 Report

Comments and Suggestions for Authors

The authors present an insightful study examining the correlation between HKA, JLO, and distal femoral rotation to determine whether the nine knee morphotypes described by Mac-Dessi are linked to specific rotational patterns. They find no significant correlation, except for valgus knees, which exhibit greater external rotation. Their findings suggest that distal femoral torsional variables cannot be inferred solely from frontal knee morphology.

Overall, the study is well-written and engaging, with clearly acknowledged limitations.

Minor Suggestions

  • Introduction: No changes needed.
  • Methods: Provide a detailed explanation of how Figures 2–6 were generated, including outlier exclusion criteria, statistical analyses performed, and the methods used to describe and analyze data distribution. Ensure that visual trends are supported by statistical inference.
  • Results:
    • In Figures 4 and 5, overlay Roman numerals corresponding to the nine knee morphotypes for better readability.
    • In Figures 4 and 5, report statistical analysis results and specify the methods used (e.g., generalized linear models) to provide numerical validation alongside visual trends in rotation distribution.

Author Response

Comments and Suggestions for Authors Reviewer 2

The authors present an insightful study examining the correlation between HKA, JLO, and distal femoral rotation to determine whether the nine knee morphotypes described by Mac-Dessi are linked to specific rotational patterns. They find no significant correlation, except for valgus knees, which exhibit greater external rotation. Their findings suggest that distal femoral torsional variables cannot be inferred solely from frontal knee morphology.

Overall, the study is well-written and engaging, with clearly acknowledged limitations.

Thank you very much for your thorough review, comments and suggestions. We will try to incorporate the changes you propose to improve the quality of the article and make it suitable for publication. Thank you very much for your kind contribution.

I will provide a point-by-point response to your comments and objections and indicate the modifications I propose to the manuscript to incorporate your comments.

Minor Suggestions

Introduction: No changes needed.

Methods: Provide a detailed explanation of how Figures 2–6 were generated, including outlier exclusion criteria, statistical analyses performed, and the methods used to describe and analyze data distribution. Ensure that visual trends are supported by statistical inference.

The exclusion of outliers is based on the criteria proposed by Tukey (1977), with values greater than Q3+ (1.5IQR) and less than Q1-(1.5IQR) being excluded.

Eliminating outliers is imperative, as they have been shown to significantly impact the mean of the various data distributions (Tukey, 1977). This subsequently affects statistical methods based on the normality assumption, which is utilised in the present study. These include confidence intervals, analysis of variance for the mean of the data distributions corresponding to each CPAK classification group, and Tukey's multiple comparisons. In addition, the normality of the distributions was verified using SPSS.

The detection of atypical data using the IQR is a statistical method that has been widely adopted and strongly supported in literature for a very long time. It is little used in medical literature and widely used in other branches of science. We include the classic references that support it:

Tukey, John W. Exploratory Data Analysis. Addison-Wesley. (1977). ISBN 0-201-07616-0.

Freedman, D., Pisani, R., Purves, R. Statistics (4th ed.). Norton & Company. (2007). ISBN 978-0393929720.

The interquartile range (IQR) is a measure of dispersion statistics employed to assess the variability in the intermediate range of a data set, excluding outliers or extremes (i.e. values that can distort the traditional range). In other words, the IQR is an indicator that allows us to approximate the variability of observations close to the median. The interquartile range is a more robust measure than the traditional range because it is based on the quartile values rather than the maximum and minimum values of the data set.

The advantages of the interquartile range include the following:

  • It is not affected by extreme or atypical values.
  • This is particularly beneficial in cases where the distribution is highly asymmetric, meaning that the majority of values do not align with the mean. In such scenarios, the interquartile range becomes a more effective measure of dispersion when compared to the range.
  • Additionally, the median is a more suitable measure of central tendency in asymmetric distributions. While the interquartile range is relatively straightforward to calculate, it is not as simple as the traditional range.
  • The result is measured in the same units as the data being analysed.

However, the following disadvantage is associated with the interquartile range:

  • It takes longer to calculate than the traditional range.

Given the study’s objective of achieving phenotypic characterisation, the exclusion of atypical values encompasses cases in which the value of aHKA, JLO, and CTA is contingent on a marked osteoarthritic pathological alteration and not on a phenotypic morphological characteristic. Consequently, excluding these atypical values will yield a more precise result.

The dispersion of data that does not conform to a perfectly symmetrical distribution cannot be summarised by a single number. Consequently, the standard deviation can only assist in distinguishing data that is a certain distance from the sample mean, subsequent to eliminating outliers, thereby rendering the distribution more normal (i.e., closer to the symmetrical distribution model followed by the data studied).

All the figures, except for Figure 6 (which was self-made using Photoshop), were generated with MiniTab v.22 software.

Results:

In Figures 4 and 5, overlay Roman numerals corresponding to the nine knee morphotypes for better readability.

Figure 5 has been removed. We have decided to eliminate it to avoid confusion. Figure 5 was only an alternative way of displaying the information in Figure 4. The heterogeneity of the values could be observed in three dimensions.

We want to express our sincere gratitude for the insightful observation regarding the numbering of the CPAK groups. We will proceed to implement the recommended numbering system as outlined.

In Figures 4 and 5, report statistical analysis results and specify the methods used (e.g., generalized linear models) to provide numerical validation alongside visual trends in rotation distribution.

As previously stated, Figure 5 has been eliminated. It should be noted that Figures 4 and 5 are merely visual summaries of the heterogeneous distribution of the CTA data in the domain of the function f(aHKA, JLO)=>CTA.

These figures are generated automatically by the statistical programme Minitab v.22 from the discrete values with which this actual function of two real variables is defined.

Thank you for your thorough review. Thank you for the ideas to improve the current manuscript. Thank you very much.

Round 2

Reviewer 1 Report

Comments and Suggestions for Authors

The revisions have significantly improved the clarity and depth of the manuscript. The explanations are clear, and the clinical relevance is well-addressed.